# Changes in management of owned cats in the countryside – A comparison of results from surveys undertaken in the same rural area of Denmark in 1998 and 2022

Peter Sandøe[1,2]*, Ulrike Gade[1], Marianne Lund Ujvári[3], Bryndis Wöhler[1], Thomas Bøker Lund[2], Henrik Meilby[2], Clare Palmer[4], Søren Saxmose Nielsen[1]

1 Department of Veterinary and Animal Sciences, University of Copenhagen, Copenhagen, Denmark,
2 Department of Food and Resource Economics, University of Copenhagen, Copenhagen, Denmark,
3 Animal Protection Denmark, Copenhagen, Denmark, 4 Department of Philosophy, Texas A&M University, College Station, Texas, United States of America

* pes@sund.ku.dk

## Abstract

This study investigates changes in the management of owned domestic cats (*Felis catus*) in a rural area of Denmark, at two points in time separated by 24 years. Households in a 47 km$^2$ area, and on 23 farms near this area, were presented with the same questionnaire in 1998 and in 2022. Additional data about the number of cats earmarked/microchipped and registered in 1998 were provided by two cat registers. The study population was compared to other rural areas in Denmark using data collected in 2021 from a nationwide probability survey of cat owners. The study area was found to represent rural areas in Denmark well. From 1998 to 2022, our study found a slight drop in the total number of owned cats. There was a major shift away from cats living on farms; this was true both of full-scale farms (that is, farms from which the farmer makes a living) where the number of cats decreased by two-thirds between 1998 and 2022 compared to 1998; and on hobby farms, where the number halved over the period. However, the number of cats living in residential homes nearly doubled, and, correspondingly, there was also a significant increase in the proportion of cats with indoor access. Cat owners have increasingly adopted measures to manage their cats, including much higher proportions of cats being spayed and castrated (there was an increase from 61% to 98% among male cats with indoor access and from 13% to 70% among male cats without indoor access). There was, also, a major increase in cats that have been earmarked/microchipped and registered, rising from 8% to 64%. This change in owner behaviour has led to many fewer unwanted kittens being born and subsequently killed. The findings therefore provide evidence of changing rural human-cat relationships, with rural cat owners increasingly managing their cats in the same way as urban cat owners. Our findings also help to explain recent evidence that the number of unowned cats in Denmark is much lower than previously assumed.

**Data availability statement:** All relevant data are within the manuscript and its Supporting Information files.

**Funding:** This study was supported by Animal Welfare Denmark and Aage V. Jensens Fonde in the form of grants awarded to MLU, by Kitty og Viggo Freisleben Jensens Fond by grants awarded to BW and UG, and by Skibsreder Per Henriksen, R. og Hustrus Fond by a grant awarded to PS. The funders had no role in study design, data collection and analysis, decision to publish, or preparation of the manuscript.

**Competing interests:** Marianne Lund Ujvári worked for Animal Protection Denmark while doing the first part of the project in the late 1990s. The remaining authors have declared that no competing interests exist. This does not alter our adherence to PLOS ONE policies on sharing data and materials.

## Introduction

Populations of unowned domestic cats (*Felis catus*) – whether unsocialized feral cats that have never lived with humans, or socialized cats that have strayed or been abandoned – are regarded as problematic in many countries [1–3]. It is argued that they are a nuisance, that they transmit disease to humans, owned cats and wildlife [4–6]; that the unowned cats themselves suffer from poor welfare [4,7,8] and that they reduce biodiversity by hunting [9]. For all these reasons, there is wide agreement that populations of unowned cats should be controlled. Strategies for doing so include rehoming socialized cats through animal shelters, trapping and neutering unsocialized cats and returning them to their place of origin [5,6,10], and trapping and killing them [7]. Because domestic cats are loved by some and vilified by others, these strategies are all controversial. For example, those who love cats think it is too extreme to kill them, while those who vilify cats object when feral cats are returned to their place of origin after sterilization.

However, these population management strategies are ineffective if there is an ongoing influx of domestic cats from the owned population into the unowned population. Studies consistently show that colonies of feral cats increase by immigration [10,11]. This means that to prevent unowned cat populations from growing, measures are needed not only to manage existing populations, but also to prevent movement from owned to unowned populations.

One way of reducing this movement is by keeping cats indoors. In some countries, a significant percentage of cats do not have outdoor access. For instance, one study found that 63% of companion cats live wholly indoors in the USA [12]. However, keeping cats wholly indoors is controversial, generating significant concerns about cat welfare [13–15]. And confining cats indoors is not widely accepted outside North America. For instance, a 2010 study found that 95% of cats in New Zealand were allowed outdoor access [16]; and across Europe, most owned cats seem to have indoor/outdoor lifestyles [17]. In Denmark, the proportion of owned cats confined indoors or with restricted outdoor access, e.g. to fenced areas that keep cats enclosed, is around 25% [18]. Given how engrained the idea of outdoor access is in Denmark and other European countries, managing cats by keeping them indoors is unlikely to succeed in preventing the flow of owned cats and their offspring into unowned populations.

Another way of staunching the flow is to embrace what is sometimes called "responsible pet ownership". In the case of cats, this involves spaying and neutering to prevent kittens being born, the insertion of microchips or use of other forms of identification such as tattooed ear-marks, and the creation of a registry of identifiable cats to facilitate the return of strays. Some states and countries are even beginning to require either sterilization, microchipping, or both. (Microchipping, for instance, has recently become a legal requirement in England [19], while the region of Flanders in Belgium [2] and the state of Western Australia now require both sterilization and microchipping [19]).

However, historically at least, voluntary uptake of these measures seems to have varied significantly across different demographics – especially between urban/suburban and rural communities. In particular, early studies from the USA found a low level of spaying and neutering of farm and household cats in rural areas [20–22]. This matters because even if reproduction is controlled in urban areas, high reproductive rates among rural cats leads to dispersal [23] across rural areas and also likely inflow to urban/suburban areas, especially where there is provision of supplemental food [24].

Recent studies from Italy, the United Kingdom and the USA however, seem to suggest that the gap between rural and urban cat management is diminishing, with higher rural uptake in terms of spaying and neutering [25–27]. If the results of these studies can be generalized to other areas and other countries, this would suggest that populations of unowned cats should be diminishing as rural cat owners adopt measures that control the growth in cat populations.

Interestingly, a recent study in Denmark found what appears to be such a drop in the number of unowned cats, with a mean density of $2 \pm 0.3$ (SE) cats per km² [28]. This number was about 80% lower than previous estimates, and also lower than one would expect based on models where feed availability is recognized as the only factor restricting the stray and feral cat population. A similar density of unowned cats was found in a near-simultaneous study undertaken in the UK [29]. These findings directly prompted the research presented in this paper. We wanted to investigate whether a reduction in unowned cat populations could be explained by changes in the ways rural cat owners, in Denmark at least, have managed and cared for their cats over time.

The best way to test this hypothesis would be to study changes in cat owner behaviour in an identified rural area over time. However, as far as we could find, no such studies exist. This led us to the current study, where we began with the findings of a previous questionnaire survey on rural cat management undertaken in 1998 [30] and (S1 File). The 1998 study involved households in a 47 km² countryside area in Western Zealand, Denmark, and on 23 additional nearby farms. We carried out a follow-up study involving the same households and farms 24 years later, in 2022. The 1998 study was only published as a report in Danish; this is the first presentation of its results to an international scholarly audience. The 1998 study did not contain information about how many cats were earmarked/microchipped and registered at that time; so for this information, we drew on two official national cat registers that were both active in 1998. In addition, the study population was compared to the target population of cat owners in rural areas in Denmark using a probability-based survey of cat owners where data was collected in 2021.

The current study, then, is the first of its kind. It documents changes in management and care of owned cats at two points in time separated by 24 years in a typical rural area. Our research questions were: 1) How has the number of owned cats and their distribution between farms and residential houses changed? 2) How has owner behaviour in terms of owned cats with indoor access vs. outdoor-only cats (these include cats usually called barn cats, that have access to farm buildings but not residential homes) changed? 3) How has the number of kittens born in the area changed? 4) How has the use of reproductive control and other means of cat population control changed? 5) Was there an increased uptake of methods of ear- and microchip marking and registering over time? 6) How representative is the studied area for rural Denmark?

## Materials and methods

### Survey studies in 1998 and 2022 – overview

The 2022 cross-sectional study was conducted as a follow-up to a cross-sectional study carried out in late winter and early spring 1998 [30] and (S1 File). The study area was identified based on the 47 km² zone in the middle of the largest Danish island, Zealand (S1 Fig), and an additional 23 active agricultural premises with production animals located a maximum of 14 km from the 47 km² zone, as used in the original study. The original data from the 1998-study do not exist, and data were therefore extracted from the study report [30] included as supporting information, S1 File. Data collection in the present study occurred from 5 May to 1 July 2022. A questionnaire was developed to replicate the data collection in 1998. However, some adaptations had to be made, for example because buildings are now used for purposes other than farming and housing. The original study questionnaire was administered in person, and this was copied in the new study: Co-author MU walked from house to house in 1998, and co-authors UG and BW assisted by a hired student assistant did the same in 2022.

## Survey studies in 1998 and 2022 – respondents

In 1998, the study area included 488 addresses in the 47 km²-zone and 23 farms, i.e. 511 properties were visited. On 28th March 2022, the Danish Address Registry (Danmarks Adresser, The Danish Agency for Data Supply and Infrastructure, Copenhagen NV, Denmark) was used to identify all addresses in the same area. An informational letter was sent to all these addresses; some were returned by the postal service with information about the reason. All returned letters and non-responders were investigated via Google Maps (Google Inc., Mountainview, California, USA), Krak (Krak Danmark, Copenhagen V, Denmark), and the Danish Central Herd Registry (Danish Veterinary and Food Administration, Glostrup, Denmark), to see whether the premises contained facilities suitable for human housing, whether production animals were currently recorded on the premises, and whether production animals had been present earlier. Addresses with no facilities suitable for human housing (e.g. a church chapel, a power transformation station, abandoned houses in decay, and premises with commercial workshops only) were excluded. Premises with multiple addresses, e.g. apartment type buildings or other houses with multiple separate households were included but kept separate. Addresses where commercial activities and households were combined were included. All the addresses that were included were then defined as households or as "other" addresses such as hotels, bed and breakfasts, golf courses etc. that potentially could host cats. Existing working farms from which the farmer makes a living were categorised as "full-scale farms", and former farms, on which residents were not economically dependent on farming, were categorised as "small-scale or hobby farms". The additional 23 addresses with farms that were included in 1998 were also included in 2022. They were re-categorised to their new use if they were no longer farms. Written informed consent was obtained from all participants in the 2022 study (see S4 File for informed consent form).

## Survey studies in 1998 and 2022 – questionnaire

An electronic questionnaire was developed for the 2022 study in SurveyXact (Rambøll Management Consulting, Aarhus N, Denmark), based on the questions asked by Ujvári [30] and (S1 File), with some additional new questions, including questions about whether cats were earmarked/microchipped and registered. The original Danish and an English translation of the 2022 survey text are available as supporting material (see S2 File for original survey text from 2022 in Danish and S3 File for English translation). The 1998-questionnaire is available in the report from 1999 [30] and (S1 File). Briefly, the following information was retrieved at each address from the respondents: type of property, number of cats by type (understood here as cats with indoor access or outdoor-only cats) and by sex (male, female, unknown), the purpose of keeping any cats (companion animal, predator, other) by cat-type, whether specific methods for controlling the number of cats were used, and if yes, which methods (hormonal treatment, castration, sterilisation, killing of adults or kittens), whether any kittens had been born in the past year, and how any kittens born were dealt with (killed, kept, sold/given away), and whether the cats were ear- or chipmarked and registered.

## Survey studies in 1998 and 2022 – data analysis

Because the raw data from the 1998 survey no longer existed (they were lost before the current collaboration started), and because some data were presented as graphs only, some of the 1998 data were extracted by a combination of measuring the percentages shown as height of bars in barplots with a ruler, combining this with the total number of cats contributing to the graphs, and using this to estimate the number of cats for each bar.

The response percentage (calculated as number of completed questionnaires divided by the total number of addresses invited to participate in the survey) was given for the 1998-survey, while it was calculated overall and by property type for the 2022-survey.

For the 2022 survey, the number of cats per cat type (outdoor-only vs. cats with indoor access) and property type was estimated. For both surveys, the number of cats overall (1998 and 2022) and by property type (2022 only) was also estimated. Since neither the 1998 nor the 2022 data reached a 100% response percentage, these estimates were adjusted to take non-responders into account. This was carried out by dividing the number of cats by the response percentage. Furthermore, the number of kittens born was estimated.

The proportions in 1998 were compared to proportions in 2022 using the prop. test-function in R [31]. A P-value below 0.05 was considered statistically significant.

### Data from the two Danish cat registers

Two national cat registers exist in Denmark: Dansk Katteregister (c/o Inges Kattehjem, Glostrup, Denmark) and Det Danske Katteregister (c/o Danish Veterinary Association, Frederiksberg, Denmark), and both were requested to provide data for the postal codes covering the study area (4100, 4174, 4330, 4360, and 4370) in 1998. Subsequently, the addresses provided that had registered cats were merged with the addresses in the study area, to determine the proportion of registered cats in the study area in 1998.

### National survey conducted in 2021 – overview

This was a cross-sectional questionnaire survey administered and carried out by Statistics Denmark (Copenhagen Ø, Denmark). A probability sampling procedure was employed involving a random selection of Danes. In total, 5,027 persons in the age interval 18–89 years were randomly drawn from the Danish Civil Registration System and invited to participate in the survey via an e-mail that had a link to an online questionnaire. Invitees that did not respond following emailed invitations were contacted by telephone or letter, and encouraged to participate. At that point, it was also possible for invitees to respond to the questionnaire via an interviewer-assisted telephone interview. Data collection was carried out in May-June 2021.

### National survey conducted in 2021 – questionnaire and register data information

The questionnaire aimed to investigate the distribution of companion animals in Denmark, how companion animals are managed, and owner-companion animal attachment. Here, we only made use of data on how people manage their cats (355 respondents), based on two questions about 1) whether the cat has indoor access, and 2) whether the cat is identifiable (by a collar or a chip) and/or registered. Respondents that had more than one cat were not asked to provide management information about all cats. Instead, they were asked to order the cats' names alphabetically and give information for the first cat only.

We also used register data information provided by Statistics Denmark about the population density of the geographical area where the cat-owning participants, hence also the cats, live. The population density variable (*Storhed)* follows the UN's guidelines for the delineation of urban areas [32]. A population density variable with three levels was constructed (1: Large city area with (>99,999 inhabitants); 2: Medium-small city area (>199 & <100,000 inhabitants); 3: Rural area or village (<200 inhabitants).

### National survey conducted in 2021 – data analysis

The percentage of cats that were outdoor only, and the percentage of earmarked/microchipped and registered cats, were reported for each population density level.

To ensure that the results were representative of Danish households, weighted percentages were reported using a weight variable that adjusts the sample so that it matches the

background population comprising approximately 3.1 million families in Denmark. Statistics Denmark constructed this weight variable based on the following family-level census information: population density; family income; number of family members; region of Denmark; housing type; dwelling size; and family type (i.e. 'single without children', 'single with children', 'couple with children', and 'couple without children').

We compared whether there were differences across the population density levels using a $\chi^2$-test and considered a P-value of 0.05 to be statistically significant. We report P-values from tests where the weight variable was not activated. But results were similar when the weight variable was activated.

### Ethics statement

The primary survey was approved by The Research Ethics Committee at the Faculties of Science and of Health and Medical Sciences at the University of Copenhagen (ReF: 504-0319/22-5000). Informed written consent was obtained from all interviewees (see S4 File with informed consent form). The consent form is included in the data repository. The representative questionnaire study was approved by the Research Ethics Committee at the Faculties of Science and of Health and Medical Sciences at the University of Copenhagen (permission no.: 504/0159/20-5000). Respondents in the survey carried out by Statistics Denmark did not give informed consent before participation. This approach is employed by Statistics Denmark in all the questionnaire studies that they conduct.

## Results

### Comparison 1998 and 2022 – respondents

A total of 488 addresses within the study area and 23 addresses outside the study area were visited in 1998. Of these 511 addresses, 467 (response percentage: 91.4%) responded with full information; two properties only provided partial information, and they were therefore excluded. For the 2022-survey, the sample size and proportion responding (stratum-specific) were: 26 (57.7%) for full-scale farms, 151 (60.9%) for small-scale/hobby farms, 273 (60.4%) for houses (incl. those with multiple households), and 24 (54.1%) for other. Overall, this corresponded to 474 possible respondents where responses were received from 285 (response percentage: 60.1%). The full data set from the 2022 study is to be found in S1 Table.

### Comparison 1998 and 2022 – number of cats and ways of keeping them

In 1998, 59.6% of the households had cats, in comparison to 53.0% of the households in 2022. Overall, 840 cats were estimated to be present in the area in 1998, and 764 cats in 2022. The number and mean number of cats by property type are shown in Table 1.

In 1998 cats lived on 269 of the properties that responded. Of these properties 117 (43%) had outdoor-only cats and 181 (67%) had cats with indoor access (note that the sum is higher than 100% because some properties had both outdoor-only cats and cats with indoor access). In 2022 there were cats on 193 of the properties that responded. Of these properties 59 (31%) had outdoor-only cats and 148 (77%) had cats with indoor access. This was a significant decrease in the proportion of outdoor-only cats from 1998 to 2022 (p = 0.006).

### Comparison of 1998 and 2022 – kittens born and management of cats

The number of kittens born is summarised in Table 2. Overall, 410 kittens were estimated to have been born in 1998 compared with 134 in 2022, i.e., a reduction to approximately one third of the previous number. The number of surplus kittens (i.e., kittens unaccounted for) also fell from 189 to 58, i.e., 31% of surplus kittens produced in the past. Almost half (66/134) of the kittens born came from small-scale/hobby farms in 2022.

**Table 1. Distribution of cats for different property types, with observed and estimated number of cats. The estimated number of cats are the observed number of cats divided by the response percentage.**

| Property type | Observed data | | | | | | Estimated cat population | | | |
|---|---|---|---|---|---|---|---|---|---|---|
| | No. of properties§ | | No. of cats | | Mean no. cats | | Response percentage | | No. of cats | |
| | 1998 | 2022 | 1998 | 2022 | 1998 | 2022 | 1998* | 2022 | 1998 | 2022 |
| Household in residential home# | 258 | 266 | 246 | 309 | 0.95 | 1.16 | 91.4 | 60.4 | 269 | 511 |
| Farm, small-scale (hobby) | 130 | 50 | 216 | 71 | 1.66 | 1.42 | 91.4 | 60.9 | 236 | 117 |
| Farm, full-scale | 79 | 28 | 306 | 55 | 3.87 | 1.96 | 91.4 | 57.7 | 335 | 95 |
| Other | 0 | 20 | 0 | 22 | 0 | 1.10 | 91.4 | 54.1 | 0 | 41 |
| Total | 467 | 364 | 768 | 457 | 1.64 | 1.26 | 91.4 | 60.1 | 840 | 764 |

§Including properties with no cats and additional properties outside the area.

#Including properties where single houses are divided into apartment type households.

*For 1998, only pooled response percentage existed, and this was therefore used for all property types.

**Table 2. Estimated kitten births by year and type of household. Only overall numbers were available for 1998.**

| Type of household | Kittens born | | Kittens kept | Kittens killed | Surplus kittens (i.e., kittens unaccounted for) | |
|---|---|---|---|---|---|---|
| | 1998 | 2022 | 2022 | 2022 | 1998 | 2022 |
| Household in residential home# | | 35 | 18 | 9 | | 8 |
| Farm, small-scale (hobby) | | 66 | 21 | 16 | | 29 |
| Farm, full-scale | | 20 | 10 | 2 | | 8 |
| Other | | 13 | 0 | 0 | | 13 |
| Total | 410 | 134 | 49 | 27 | 189 | 58 |

#Including properties where single houses are divided in apartment type households.

The use of different types of reproductive control used in each year for the two types of cats are summarized in Table 3 with major changes seen for all types of control. Use of birth control hormones has mostly stopped, killing of adult cats and kittens has dropped a lot, whereas sterilizing of both male and female cats have increased substantially for both outdoor-only cats and cats with indoor access.

## Comparison of 1998 and 2022 – number of cats earmarked/microchipped and registered

By comparing information from the two Danish cat registers about addresses with postcodes overlapping with the area studied and the actual addresses in the area, it was found that 41 out of 511 addresses in 1998 had a cat that was earmarked/microchipped and registered, a proportion of 8%.

In the 2022 survey in total 123 (74%) of the 167 respondents with cats who responded to the question (6 respondents answered "do not know") said all or some of their cats were earmarked/microchipped and registered. On properties where cats had indoor access 78% (95/122) answered that their cats were earmarked/microchipped and registered; for those with outdoor-only cats it was 59% (35/59) (note that a few properties had both cats with indoor access and outdoor-only cats).

## Comparison of findings from study area 2022 with national survey from 2021

In total, 2,347 of the 5,027 invited respondents completed the 2021 national survey (response percentage 47%). Of these, 355 reported that there was at least one cat in the family. Eight

Table 3. Types of regulation of the cat populations in 1998 and 2022, stratified by cat type.

| Cat type | Category | 1998 | | | 2022 | | | P-value[*] |
|---|---|---|---|---|---|---|---|---|
| | | N[*] | n | % | N | n | % | |
| Access indoor | Birth control hormones | 104 | 40 | 38 | 93 | 4 | 4 | <0.0001 |
| | Sterilised female cat | 104 | 46 | 44 | 93 | 86 | 92 | <0.0001 |
| | Castrated male cat | 121 | 74 | 61 | 85 | 83 | 98 | <0.0001 |
| | Killing adult cats | 104 | 5 | 5 | 137 | 1 | 0.7 | 0.11 |
| | Killing kittens | 104 | 22 | 21 | 137 | 2 | 1 | <0.0001 |
| Outdoor only | Birth control hormones | 83 | 26 | 31 | 43 | 3 | 7 | 0.004 |
| | Sterilised female cat | 83 | 16 | 19 | 43 | 27 | 63 | <0.0001 |
| | Castrated male cat | 89 | 12 | 13 | 37 | 26 | 70 | <0.0001 |
| | Killing adult cats | 83 | 13 | 16 | 59 | 3 | 5 | 0.09 |
| | Killing kittens | 83 | 43 | 51 | 59 | 5 | 8 | <0.0001 |

[*]N is the total number of responders with the specific type of cats in the stratum, and n is the number of responders practicing the specific type of regulation. N is not the same for the different types of control measures, as not all responders had both male and female cats. The P-value indicates whether there is a difference between the practice in 1998 and 2022 based on the Z-test.

respondents were removed due to missing register data information about population density, giving a sample size of 348. This sample, however, was also reduced slightly on a case-by-case basis in the tables below, because respondents that answered "do not know" were removed.

In Table 4 the share of outdoor-only cats in the study area is compared to differently populated areas according to the national survey, to see whether the study area is like other rural areas in Denmark in this regard. As expected, outdoor-only cats were much more frequent in rural areas compared to more densely populated areas in Denmark($\chi^2$ test: p < 0.001). The share of outdoor-only cats in the study area (31%) is very similar to other rural areas (32%), and the difference is not statistically significant (one sample binomial test: p = 0.37).

Finally, in Table 5 we compared the relative number of earmarked/microchipped and/or registered cats in differently populated areas according to the national survey with the share of these cats in the study area. The share of earmarked/microchipped registered cats is lower in rural areas compared to more densely populated areas in Denmark ($\chi^2$ test: p < 0.001). In line with this, the share of earmarked/microchipped cats in the study area (64%) is lower than the shares observed in medium/small city areas (82%; one sample binomial test: p < 0.001)

Table 4. Percentage of outdoor-only cats stratified by population density from the 2021 national survey (unweighted n = 344*) and compared with the study area.

| | | %** |
|---|---|---|
| National survey | Large city area (>99,999 inhabitants) | 1 |
| | Medium-small city area (>200 & <100,000 inhabitants) | 5 |
| | Rural area or village (<200 inhabitants) | 32 |
| | Total | 12 |
| Study area | | 31 |

Weighted percentages are reported in the national survey.

*The sample size was reduced from 348 to 344 because four respondents that answered "do not know" were removed from the analysis.

**The question formulation in the national survey was: "Is the cat able to go outdoors?". The relevant response option was "Yes, and the cat is seldom or never indoors".

**Table 5. Percentage of earmarked/microchipped and/or registered cats stratified by population density from the 2021 national survey (unweighted n = 327\*) and compared with the study area.**

| | | %** |
|---|---|---|
| National survey | Large city area (>99,999 inhabitants) | 83 |
| | Medium-small city area (>200 & < 100,000 inhabitants) | 82 |
| | Rural area or village (<200 inhabitants) | 49 |
| | Total | 72 |
| Study area | | 64 |

Weighted percentages are reported in the national survey.

*The sample size was reduced from 348 to 327 because 21 respondents that answered "do not know" were removed from the analysis.

**The question formulation in the national survey was: "Is the cat marked?". The following response option were considered as a earmarked/microchipped and registered cat: "Ear marked and registered", and "Microchipped and registered".

and large city areas (83%; one sample binomial test: p < 0.001). On the other hand, the share of earmarked/microchipped and registered cats in the study area is higher than in other rural areas in Denmark (49%; one sample binomial test: p < 0.001).

## Discussion

### Outline of main findings

The study concerns cat ownership in a typical Danish rural area. It shows that there has been a slight drop in the number of owned cats and a major shift away from cats living on full-scale farms (under a third in 2022 compared to 1998) and on hobby farms (the number halved in 2022 compared to 1998). However, there has been nearly a doubling of the number of cats living in residential homes. This shift clearly reflects a drop in the number of full-scale farms in Denmark during the period (cf. Table 1). According to statistics from Statistics Denmark [33], the number of full-scale farms has dropped from 25,549 in 1998 to 7,598 in 2021. Likely as a consequence of this structural change the number of outdoor-only cats has gone down relative to the number of cats living as companion cats with indoor access, which has gone up.

The number of kittens born and the number of kittens not accounted for has gone down to a third of the original number from 1998 to 2022. This could be an important part of the explanation of the apparent drop in the number of unowned cats in Denmark.

This change in the number of kittens born may reflect shifts in methods of reproductive control used and other means of cat population control. We found important changes in the use of birth control hormones, spay and neuter, and killing. Use of birth control hormones ("the pill") has gone down dramatically, probably reflecting that veterinary guidelines no longer recommend birth control hormones as a means of contraception in cats [34]. On the other hand, the use of permanent surgical fertility control measures has gone up dramatically. Finally, killing as a means of population control among adult cats has gone down significantly, both among cats with indoor access and outdoor-only cats. The same is true for kittens. This is clearly a major shift towards more responsible cat ownership and appears to suggest a growing aversion towards killing cats as a means of population control. A dramatic increase in ear-marking, microchipping and registration of cats was also observed, both for cats with indoor access and outdoor-only cats. This too indicates a shift towards responsible cat ownership.

Compared with a recent survey covering all of Denmark, the area studied seems to fall within the rural segment, with the same level of outdoor-only cats compared to the average

rural group, but with more cats earmarked/microchipped and registered compared to the average rural area.

Overall, our study shows that over the past two and a half decades there has been a major shift in how rural cats are managed in Denmark. A higher proportion of cats have indoor access; management practices including surgical reproductive control and the marking and registering of owned cats is becoming the norm; and traditional outdoor-only owned cats are slowly reducing in numbers. Rural inhabitants are now also much less likely to kill cats. These developments, we speculate, may reflect a change in attitude to cats, where in rural as well as in urban areas they are increasingly viewed as companions or even as family members.

A recent study comparing owner attachment to cats relative to dogs in three countries, Austria, Denmark and the United Kingdom [35], shows large cross-country variation. Owners in Denmark cared less about their cats, in comparison to their dogs, than in the two other countries, Austria and the UK. This indicates that attachment to cats is not static, and even though attachment to cats is lower in Denmark compared to other countries, it most likely has increased over time. This increased attachment to cats and the related concern about cat welfare would also help to explain the relatively small population of stray and feral cats found in a recent study [28].

There have been two potentially relevant changes to Danish legislation regarding cats during the 24-year period between the two studies at issue here. Both changes apply to the law regulating confinement of animals on an owner's property, a law mainly focused on farm animals, but also covering cats. A 2014 change in the law banned the shooting of stray cats [36]; and a 2021 change in the law meant that ownership of cats that lack earmarks or microchips cannot be claimed. [37]. Both changes likely reflect the increasing perceived value of cats as companion animals.

## Comparison with previous studies

No other studies have attempted to do what we have done here. All existing studies of the management of rural cats in the Global North are cross-sectional studies undertaken at a specific point in time. More responsible cat management practices are, however, found in the most recent studies compared to the earlier studies, which is in line with the findings of the present study.

## Limitations

Our use of the early study from 1998 was affected by three limitations. Firstly, the original dataset no longer exists, so that some of the data had to be reconstructed based on inspection of graphs. Secondly, the original questionnaire had some limitations – for example, it was not possible to add up the number of cats across the two main categories, "with indoor access" and "outdoor only". Thirdly, there were no real measures of people's attachment to their cats.

The 2022 study was limited by the lower response rate of 60% compared to 91% in the early study. However, by normal standards, 60% is still a very high response rate. In the later study it appears that some possible respondents may have been missed due to letters being returned by the postal service, even though there were people living at some of the addresses.

Also it is a limitation that number of cats being earmarked/microchipped and registered were measured in different ways in 1998 and 2022, with data from cat registries in 1998 and with survey-based data in 2022. However, given the large difference in numbers, this limitation is likely not very important.

## Conclusions

The study provides evidence of changing rural human-cat relationships, with rural cat owners increasingly managing cats in the same way as urban cat owners. This probably reflects both

a diminishing divide between rural and urban life in a small and highly industrialised country like Denmark and a growing human attachment to owned cats. The study also helps to explain why the number of unowned cats in Denmark recently has been found to be much lower than was previously assumed.

## Supporting information

**S1 File. The original report, Ujvari, 1999.**
(PDF)

**S2 File. Original survey text from 2022 in Danish.**
(PDF)

**S3 File. English translation of survey text from 2022.**
(PDF)

**S4 File. Informed consent form from 2022.**
(PDF)

**S1 Table. Data set from the 2022 study.**
(XLSX)

**S1 Fig. Map of the area.** Based on map data provided by The Danish Agency for Climate Data, GeoDanmark, which has granted users permission to use the data under the CC BY 4.0 license.
(TIF)

## Acknowledgments

We want to thank the respondents for their participation in the surveys on which this presentation is based. We also want to thank Jørgen S. Petersen and Per Hvenegaard for information on cat registration in 1998. Also we owe thanks to Eliza Ruiz Izaguirre for assistance in searching literature and to Nicoline Skandov for assisting UG and BW with the data collection in 2022. Finally, we want to thank two anonymous referees for extremely perceptive comments that helped to improve the quality of the paper in numerous ways.

## Author contributions

**Conceptualization:** Peter Sandøe, Ulrike Gade, Marianne Lund Ujvári, Bryndis Wöhler, Henrik Meilby, Søren Saxmose Nielsen.

**Data curation:** Ulrike Gade, Marianne Lund Ujvári, Bryndis Wöhler, Thomas Bøker Lund, Henrik Meilby, Søren Saxmose Nielsen.

**Formal analysis:** Ulrike Gade, Marianne Lund Ujvári, Bryndis Wöhler, Thomas Bøker Lund, Henrik Meilby, Søren Saxmose Nielsen.

**Funding acquisition:** Peter Sandøe.

**Investigation:** Ulrike Gade, Marianne Lund Ujvári, Bryndis Wöhler.

**Methodology:** Peter Sandøe, Ulrike Gade, Marianne Lund Ujvári, Bryndis Wöhler, Thomas Bøker Lund, Henrik Meilby, Søren Saxmose Nielsen.

**Project administration:** Peter Sandøe.

**Supervision:** Peter Sandøe, Søren Saxmose Nielsen.

**Visualization:** Søren Saxmose Nielsen.

**Writing – original draft:** Peter Sandøe, Søren Saxmose Nielsen.

**Writing – review & editing:** Peter Sandøe, Ulrike Gade, Marianne Lund Ujvári, Bryndis Wöhler, Thomas Bøker Lund, Henrik Meilby, Clare Palmer, Søren Saxmose Nielsen.

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
