## [Decision Letter · Decision Letter 0]

29 Oct 2024

PONE-D-24-41843Change in management of cats in the countryside – A comparison of results from surveys undertaken in the same rural area of Denmark in 1998 and 2022PLOS ONE Dear Dr. Sandøe,

Thank you for submitting your manuscript to PLOS ONE. After careful consideration, we feel that it has merit but does not fully meet PLOS ONE’s publication criteria as it currently stands. Therefore, we invite you to submit a revised version of the manuscript that addresses the points raised during the review process.

Please submit your revised manuscript by Dec 13 2024 11:59PM. If you will need more time than this to complete your revisions, please reply to this message or contact the journal office at plosone@plos.org . Please include the following items when submitting your revised manuscript:

We look forward to receiving your revised manuscript.

Kind regards,

Joshua Kamani, PhD

Academic Editor

PLOS ONE

“We want to thank the respondents for their participation in the surveys on which this presentation is based. We also want to thank Jørgen S. Petersen and Per Hvenegaard for information on cat registration in 1998. Also we owe thanks to Eliza Ruiz Izaguirre for assistance in search of literature and to Nicoline Skandov for assisting UG and BW with the data collection in 2022. Finally, thanks are due to Skibsreder Per Henriksen, R. og Hustrus Fond and Kitty og Viggo Freisleben Jensens Fond for economic support of the 2022 study and Animal Welfare Denmark and Aage V. Jensens Fonde for economic support of the 1998 study. The funders had no influence on the design of the study and the interpretation of the presented results.”

“We received funding from Skibsreder Per Henriksen, R. og Hustrus Fond and Kitty og Viggo Freisleben Jensens Fond for the 2022 study and from Animal Welfare Denmark and Aage V. Jensens Fonde for  the 1998 study. The funders had no influence on the design of the study and the interpretation of the presented results.”

“Marianne Lund Ujvári worked for Animal Protection Denmark while doing the first part of the project in the late 1990ies. The remaining authors have declared that no competing interests exist.”

4. We note that Figure S1 (Map Country cats paper.pdf ) in your submission contain [map/satellite] images which may be copyrighted. All PLOS content is published under the Creative Commons Attribution License (CC BY 4.0), which means that the manuscript, images, and Supporting Information files will be freely available online, and any third party is permitted to access, download, copy, distribute, and use these materials in any way, even commercially, with proper attribution. For these reasons, we cannot publish previously copyrighted maps or satellite images created using proprietary data, such as Google software (Google Maps, Street View, and Earth). For more information, see our copyright guidelines: http://journals.plos.org/plosone/s/licenses-and-copyright.

a. You may seek permission from the original copyright holder of Figure S1 to publish the content specifically under the CC BY 4.0 license. 

Additional Editor Comments:

Dear Authors

Kindly address all the concerns raised by the reviewers of this manuscript.

Reviewers' comments:

Reviewer's Responses to Questions

**Comments to the Author**

1. Is the manuscript technically sound, and do the data support the conclusions?

Reviewer #1: Yes

Reviewer #2: Yes

2. Has the statistical analysis been performed appropriately and rigorously? 

Reviewer #1: Yes

Reviewer #2: Yes

3. Have the authors made all data underlying the findings in their manuscript fully available?

Reviewer #1: Yes

Reviewer #2: Yes

4. Is the manuscript presented in an intelligible fashion and written in standard English?

Reviewer #1: No

Reviewer #2: Yes

5. Review Comments to the Author

Reviewer #1: Dear authors,

The paper is very interesting and important evidence for a shift in household cat keeping.

My suggestions are for your inspiration.

There are some minor issues to have a look at, according to me, this would help for clarifying some things I missed.

- Terminology: be consistent in terminology: (1) euthanasia or killing, and if different define so. (2) marking of cats, defining in the beginning what you mean, would help the reader (3) the most important one: be consistent in your categories or defining of cats (household, owned, domestic, colonies,...)! (see suggestions in pdf)

- Some more specific numbers and references to papers used would help the reader (see suggestions in pdf).

- Add the complete survey as an appendix, for me this is very important as a reader.

- Maybe a more precise title would help the reader from the onset of reading

- Management by owners and production of kittens seemed strangely (and not respectfully for me) worded.

For me as a reader it would have been helpful if certain legislative changes (in relation to cats) or other societal changes in Denmark would have been described and may have helped in understanding the trend you described. I think there is no need to state (in lines 432-434) because of not using measures on attitudes, to try do give a hint what could be influential. ("Thirdly, there were no real measures of people’s attachment to 433 their cats and therefore efforts to explain the findings in terms of changes in attitudes to 434 owned cats are bound to be speculative.") Ofcourse this will be speculative, but there should be literature on this, that has been trying to find out, why cat keeping has changed over time. I would also be really interested into reading what could be important studies to help to find this out in Denmark.

I wish you a good publication,

Best regards

Ciska

Reviewer #2: A great read and a welcome consideration of changing trends in cat ownership. I would welcome a version of this manuscript in publication.

I was unable to review the questionnaires as they were not available in English. Apologies, it may be that some of my questions would have been addressed by having access to these, if this is the case, please take my comments to indicate further information in the main text would be helpful.

The version of the questionnaire translated into English appended as supplementary material as indicated L181 would be helpful.

Also, my main comments are around improving the clarity of the manuscript specifically around cat terminology used throughout.

Main comments

Abstract and throughout- no definition of what a “full-scale” farm is compared to a “hobby” farm. Please clarify for the reader

Marked and registered is used throughout but unclear of exactly what this means- assume microchipping, while wearing a collar is marking but not registering. So it’s important to define what actually is being compared. Also along these lines:

• L86 says “other identifying marks” and I am unsure what this means.

• Terminology used inconsistently throughout the manuscript there are instances of “marked and registered”, “marked or registered”, “marked/registered” and “marked and/or registered”. These could all mean slightly different things can this be clearly defined and used consistently throughout. Especially in results where it is unclear whether like for like scenarios are being compared e.g. L321 “in 1998 had a cat that was marked and registered” and for 2022 L325 “On properties where cats had indoor access 78% (95/122) answered 326 that cats were marked or registered” So you would expect different % as the text indicates different things are been considered in 1998 and 2022.

The first paragraph of the introduction is largely repetitive of the abstract/conclusion. Suggest remove.

L17 and throughout the authors use “over a 24 year time period” this implies cat populations were followed during this period. Suggest rephrasing for clarity e.g. following a 24 year time period or pre and post a 24 year time period.

The cat terminology used is confusing and ambiguity in terminology is a potential limitation that should be mentioned in the discussion. E.g.

• “barn cats” are mentioned but not defined (e.g. L131, L393, L413) and unclear whether these are pet cats or all cats in rural areas? Be good to compare to how cats are defined in the international literature

• The authors categorise by access to indoors and not access to indoors, could the latter category include cats not considered owned?

• Does the questionnaire ask about all cats, i.e. pet, farm, feral may be thought of as different so if only considering owned cats that does not mean a drop in cat numbers

• Also, there is no consideration of indoor-only cats.

Unclear in the methods the purpose of the information letter (L155) were these completed and returned by residents or are the returns mentioned due to the postal service being unable to deliver?

L257 “the weight variable was not activated” why? this is a bit unclear, could this potentially be a limitation to mention?

The Discussion is reported in an unusual way under a series of question headings, also it is largely repetitive of the results, Here I would expect to find prose and wider implications, bringing findings together to paint an overall picture, limitations, suggestions for future studies and certainly a conclusion.

Minor comments

L25 “significant drop in the proportion of cats without indoor access” This double negative is confusing, suggest rephrase e.g. increase in cats with indoor access

L89 Microchipping is not legal in all countries of the UK, just England, suggest reword

L124 “In addition, the study population was compared to the target population of rural areas” unclear on what this means? Target population from the national cat registers?

L128 introduce the scientific name at point of first use and also as a domestic cat.

Couldn’t see any explanatory caption for Figure S1

L208 and L257 “p-value of 0.05”, should be below 0.05

L241-242 specifies a population density but no unit of area is provided, please clarify.

L371 website should be referenced appropriately with access date

L380 typo surplus. Also, unclear of the definition of surplus, would not euthanised in some circumstances be surplus or is this known to be on health grounds?

L398 killing is population control not specifically reproduction

L446 “However, overall the limitations are not seen as major.” This is subjective and should be removed and as mentioned previously a conclusion paragraph would be great.

6. PLOS authors have the option to publish the peer review history of their article (what does this mean? ). If published, this will include your full peer review and any attached files.

**Do you want your identity to be public for this peer review?** For information about this choice, including consent withdrawal, please see our Privacy Policy .

Reviewer #1: No

Reviewer #2: No

---

## [Author Response · Author response to Decision Letter 1]

16 Nov 2024

See uploaded file with response to editor and reviewers.

---

## [Decision Letter · Decision Letter 1]

8 Dec 2024

PONE-D-24-41843R1Changes in management of owned cats in the countryside – A comparison of results from surveys undertaken in the same rural area of Denmark in 1998 and 2022PLOS ONE

Dear Dr. Sandøe,

Thank you for submitting your manuscript to PLOS ONE. After careful consideration, we feel that it has merit but does not fully meet PLOS ONE’s publication criteria as it currently stands. Therefore, we invite you to submit a revised version of the manuscript that addresses the points raised during the review process.

We look forward to receiving your revised manuscript.

Kind regards,

Joshua Kamani, PhD

Academic Editor

PLOS ONE

Journal Requirements:

Reviewers' comments:

Reviewer's Responses to Questions

**Comments to the Author**

1. If the authors have adequately addressed your comments raised in a previous round of review and you feel that this manuscript is now acceptable for publication, you may indicate that here to bypass the “Comments to the Author” section, enter your conflict of interest statement in the “Confidential to Editor” section, and submit your "Accept" recommendation.

Reviewer #1: All comments have been addressed

2. Is the manuscript technically sound, and do the data support the conclusions?

Reviewer #1: Yes

3. Has the statistical analysis been performed appropriately and rigorously? 

Reviewer #1: Yes

4. Have the authors made all data underlying the findings in their manuscript fully available?

Reviewer #1: Yes

5. Is the manuscript presented in an intelligible fashion and written in standard English?

Reviewer #1: No

6. Review Comments to the Author

Reviewer #1: Dear authors,

I want to click "accept," but I held off due to one final, really minor but important concern regarding the language. Please review the text to ensure it is polished and suitable for publication—for example, replacing contractions like "don't" with "do not."

Thank you for your interesting study! All the clarifications and added material were very useful for me.

Best regards

7. PLOS authors have the option to publish the peer review history of their article (what does this mean? ). If published, this will include your full peer review and any attached files.

**Do you want your identity to be public for this peer review?** For information about this choice, including consent withdrawal, please see our Privacy Policy .

Reviewer #1: No

---

## [Author Response · Author response to Decision Letter 2]

14 Dec 2024

We received the following single comment:

Reviewer #1: Dear authors,

I want to click "accept," but I held off due to one final, really minor but important concern regarding the language. Please review the text to ensure it is polished and suitable for publication—for example, replacing contractions like "don't" with "do not."

Thank you for your interesting study! All the clarifications and added material were very useful for me.

Best regards

Our response:

Thank you for the kind words and all the good comments that have helped us improve our paper a lot. Now the text has been carefully reviewed by the native English speaker among the authors, and among other things all contractions have been replaced.

---

## [Editor Report · Decision Letter 2]

16 Dec 2024

Changes in management of owned cats in the countryside – A comparison of results from surveys undertaken in the same rural area of Denmark in 1998 and 2022

PONE-D-24-41843R2

Dear Dr. Sandøe,

We’re pleased to inform you that your manuscript has been judged scientifically suitable for publication and will be formally accepted for publication once it meets all outstanding technical requirements.

Kind regards,

Joshua Kamani, PhD

Academic Editor

PLOS ONE
---

## [Editor Report · Acceptance letter]

PONE-D-24-41843R2

PLOS ONE

Dear Dr. Sandøe,

I'm pleased to inform you that your manuscript has been deemed suitable for publication in PLOS ONE. Congratulations! Your manuscript is now being handed over to our production team.

Kind regards,

on behalf of

Dr. Joshua Kamani

Academic Editor

PLOS ONE